# BIO2TOKEN: ALL-ATOM TOKENIZATION OF ANY BIOMOLECULAR STRUCTURE WITH MAMBA

## ABSTRACT

Efficient encoding and representation of large 3D molecular structures with high fidelity is critical for biomolecular design applications. Despite this, many representation learning approaches restrict themselves to modeling smaller systems or use coarse-grained approximations of the systems, for example modeling proteins at the resolution of amino acid residues rather than at the level of individual atoms. To address this, we develop quantized auto-encoders that learn atom-level tokenizations of complete proteins, RNA and small molecule structures with reconstruction accuracies well below 1 Angstrom. We demonstrate that a simple Mamba state space model architecture is efficient compared to an SE(3)-invariant IPA architecture, reaches competitive accuracies and can scale to systems with almost 100,000 atoms. The learned structure tokens of bio2token may serve as the input for all-atom generative models in the future.

## 1 INTRODUCTION

**Background.** Biomolecular structures can be represented as 3D point clouds, where each point corresponds to a chemical entity such as an atom, a functional group, or a set of atoms that make up larger molecular entities like side-chains and nucleotide bases. Generative modeling of these structures, especially for large biomolecules, often employs coarse-grained representations to manage complexity. Popular machine learning methods include denoising diffusion probabilistic models (DDPMs) and language models. Diffusion approaches generate 3D structures and features at varying levels of detail, from individual atoms to residues. Notably, DDPMs have been used to generate atomistic conformers of small molecules (Hoogeboom et al., 2022) and design ligands for protein pockets (Schneuing et al., 2022). Applying denoising diffusion at atomistic resolution to large structures like proteins is computationally challenging. Recent models like RFDiffusion- All-atom mitigate this by diffusing only the four-atom protein backbone and reconstructing side-chains with a separate inverse folding model (Krishna et al., 2024; Dauparas et al., 2022). Similarly, language model approaches such as ESM-3 (Hayes et al., 2024) rely on coarse-grained representations at the residue level for their core modules. Despite this coarse-graining, the capacity of these models to process large proteins and protein-ligand complexes is still limited.

An effective atomic-resolution representation model for large molecules would have to be able to reason over long-range data inputs. This is because many atoms can have critical interactions with atoms that are distal in linear sequence. However, scaling traditional "workhorse" architectures such as transformers and graph-based models to accommodate such long sequences has been challenging. Here, we leverage Mamba (Gu & Dao, 2023), a selective structured state space model, to replace transformer modules in structure tokenizer models and increase the resolution to all atoms. Mamba has been developed for long-context modeling and has been demonstrated to efficiently model tasks with thousands to millions of tokens on conventional GPU hardware.

**3D structure tokenization for generative modeling.** Turning 3D structures into discrete 1D sequences for generative language modeling, discrete diffusion or other downstream task has become a popular approach to biomolecular modeling. FoldSeek

introduced the "3Di" structural interaction alphabet to convert three-dimensional protein backbone structures into one-dimensional sequences, facilitating faster structural alignment (van Kempen et al., 2022). Neural network-based quantized auto-encoders (QAEs) (Van Den Oord et al., 2017) have since been employed to learn 3D structure tokenizers. ESM-3 utilizes a transformer-based QAE that encodes residue-level backbones and decodes to all-atom structures, with training limited to proteins with fewer than 512 residues and using a 600M parameters transformer model. FoldToken (Gao et al., 2024) and InstaDeep (Gaujac et al., 2024) also use QAEs with transformer and graph neural network architectures, respectively, focusing on residue-level tokenization but limited to backbone reconstruction. Alphafold-3 (AF-3) (Abramson et al., 2024) generates all-atom structures using a token-guided diffusion network. For small molecules, approaches include one-hot encoding of coordinate digit strings (Flam-Shepherd & Aspuru-Guzik, 2023; Zholus et al., 2024) and SE(3)-invariant QAEs like Geo2Seq (Li et al., 2024) and MolStructTok (Anonymous, 2024). Prior work predominantly relies on QAEs with various architectures and features, incorporating symmetries through structural features or invariant point attention. In contrast, our method uses neither engineered SE(3)-invariant features nor does it employ invariant network architectures.

**What if we could efficiently encode and decode any biomolecule at the all-atom level?**
In this work we present:

(i) A Mamba based quantized auto-encoder for all-atom biomolecular structures that tokenizes 3D point clouds into 1D discrete tokens. We train small molecule-only, protein-only, and RNA-only vocabularies *mol2token, protein2token* and *rna2token*. We also train a unified tokenizer *bio2token* that encodes any of those biomolecules, ranging from tens to tens of thousands of atoms, that would be challenging for transformer-based methods to scale too.

(ii) A simple and compute efficient approach towards all-atom structure tokenization that does not use SE(3) invariances. Our tokenizer models are lightweight in size (1.2M parameters), with fast training and inference.

## 1.1 Background: Transformers, State space models, and Mamba

**Transformer.** Transformers (Vaswani, 2017) use the *attention* mechanism to capture long-range dependencies in sequences. Intuitively, the attention mechanism can be thought of as a fully-connected graph neural network, with the update rule:

$$y = M(x)x, \tag{1}$$

where $x$ is the input sequence, $y$ is the latent representation, and $M(x) =$ softmax$\left(Q(x)K(x)^T\right)$ is the attention matrix, or analogously adjacency matrix. The matrix multiplication formulation of attention makes it ideal for GPU processing, and has thus made transformers the workhorse of sequence modeling. However, due to the fully connected graph structure, transformers suffer from $O(N^2)$ compute and memory costs with respect to sequence length $N$.

**Mamba.** Recent alternatives such as deep structured state space models (SSM) (Gu et al., 2021; Gu & Dao, 2023; Dao & Gu, 2024) have gained traction in the field of sequence modeling thanks to their ability to overcome the quadratic bottleneck and scale to extremely long context lengths. The basic linear time-invariant (LTI) SSM is a linear recurrent neural network (RNN) with the update rule:

$$h_t = Ah_{t-1} + Bx_t, \quad y_t = Ch_t, \tag{2}$$

where $x$ and $y$ are the input and output sequences, respectively, $h$ is the RNN state, and $A, B, C$ are learnable parameters. This linear recurrence can be unrolled across time and computed equivalently as a convolution $y = K * x$, where $K = \left(CB, CAB, ..., CA^{N-1}B\right)$. This makes SSMs parallelizable over sequence positions, incurring cost $O(N \log N)$.

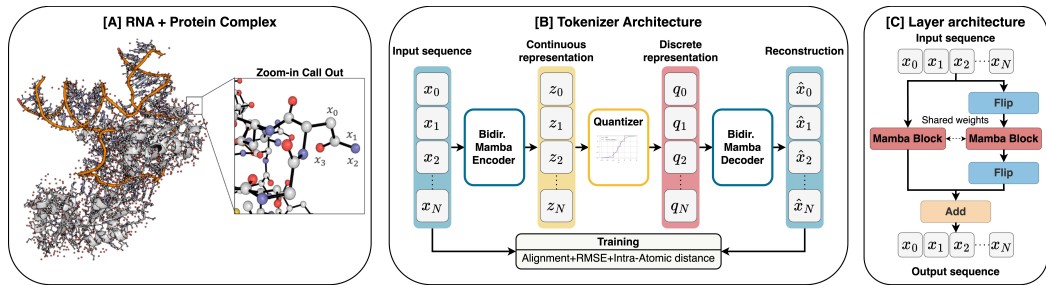

Figure 1: [**A**] Biomolecular system of interest with many thousands of atoms, with a magnified section with annotations for specific points in the point cloud. [**B**] Illustration of our tokenizer model, transforming point clouds into tokens and then back to point clouds. [**C**] Implementation details of the bidirectional Mamba layer. The first branch processes the original input using a Mamba block, while the second branch handles the flipped version of the input, reversing the output back to its original orientation afterward. The final step involves adding the results of both branches together. Notably, the two Mamba blocks in the branches share the same weights.

## 2 METHODS

Our structure tokenizer model is a QAE and Fig. 1 provides an overview of the process. The steps can be broken down into:

(i) Representation of biomolecular systems as 3D point clouds

(ii) Encoding atom positions into latent vectors

(iii) Quantizing these latent vectors into tokens

(iv) Decoding these tokens back into a 3D point cloud

**Tokenization of 3D point clouds.** A biomolecular structure $X$ of $N$ heavy atoms is represented as a $X \in \mathbb{R}^{N \times 3}$ point cloud. This point cloud is atom identity agnostic and carries no information about residue or atom type. In general, AEs compress the input to a downsampled latent space representation $Z$, where $Z \in \mathbb{R}^{n \times d}$, with $n < N$, and reconstruct the original input according to a loss function $L(X, \tilde{X})$. The first module of the AE is the encoder network $enc_\theta(X) = Z$, the second module is a decoder network $dec_\psi(Z) = \tilde{X}$. In this work, we want to maintain the all-atom resolution, and $enc_\theta(X)$ does not compress the input: $n = N$. It can be regarded as a transformation of the input into a lossless latent space. The compression in our QAE happens in the subsequent quantization of $Z$ into discrete values $Q$ as the purpose of a tokenizer is to map continuous data to discrete unique IDs, defined by a vocabulary. The latent space of a "perfect" lossless auto-encoder ($L(X, \tilde{X}) = 0$) is a perfect tokenizer if that latent space was discrete. This is usually achieved through a quantization network $quant_\phi(\tilde{Z})$, otherwise known as a tokenizer model. This tokenizer model can be optimized with the same reconstruction loss $L(X, \tilde{X})$. This tokenizer model keeps a one-to-one correspondence between the positions of input atoms, tokens and reconstructed atoms.

**Quantization.** Quantization networks learn discrete codebooks, in other words a vocabulary, of the training data. A common approach is vector-quantization (VQ) (Gray, 1984), e.g. ESM-3's tokenizer is VQ-based. VQ is notoriously hard to optimize and tends to suffer from codebook collapse. In this paper, we use a recently proposed simplified alternative, Finite-Scalar Quantization (FSQ), which produces a more efficient coverage of the codebook (Mentzer et al., 2023). FSQ has been shown to be easier to train, whilst maintaining similar levels of expressiveness as VQ-VAEs. FSQ projects the input into a hypercube of integer length $L$ and dimensions $D$ (where $D < 8$ usually). The projected input point in the hypercube is then rounded to the nearest integer set $\{0, 1, ..., L\}$. The final code/token is the product of all integer coordinates in the hypercube.

**Losses.** The ground truth and the decoded point clouds $X$ and $\tilde{X}$ are aligned via Umeyama-Kabsch algorithm (Lawrence et al., 2019) and use the total structure RMSE loss:

$$L_{\text{RMSE}}(X, \tilde{X}) = \sqrt{\frac{1}{n} \sum_{i=1}^{n} \|\mathbf{x}_i - \tilde{\mathbf{x}}_i\|^2}$$

Additionally we use an inter-atomic distance loss, that calculates the difference between the ground truth and reconstructed pairwise distances between each atom within a residue $r$:

$$\text{Loss}_{atom-dist} = \sqrt{\sum_r \sum_{i \in \mathcal{R}_r} \sum_{\substack{j \in \mathcal{R}_r \\ j \neq i}} \left( \|x_i - x_j\| - \|\tilde{x}_i - \tilde{x}_j\| \right)^2} \tag{3}$$

where $\mathcal{R}_r$ is the set of atom indices in $r$. In the case of small molecules, this is calculated over the entire molecule. The total loss for optimization is equally weighted between total structure RMSE and inter-atomic distance loss.

## 3 Experimental Details

### 3.1 Datasets

An overview of all training and test data is provided in the Appendix table 3. In a pre-processing step all point clouds of the biomolecular structures are translated to be centered at the zero-origin.

**Small molecules:** Small molecules, typically organic molecules below a 500 Dalton weight, are not static. At standard temperatures and pressures they take on various 3D structural conformations, each having a specific conformational energy. We used the $\nabla^2$DFT dataset (Khrabrov et al., 2024) of 1.9M small molecules with a total of 16M simulated structural conformations as a source of data. This dataset provides train and test splits for multiple levels of generalizability: a *test-conformer* split of unseen conformations of molecules, a *test-structure* split of unseen molecules and all their conformations, and a *test-scaffold* split of unseen scaffold classes of molecules and their conformations. The minimum number of heavy atoms in this dataset is 8 and the maximum is 27.

**Proteins:** We prototype and run various hyperparameter studies on the CATH 4.2 dataset of 18k protein structures from the PDB, with train-test splits on the CATH topology classifications as defined by Ingraham et al. (2019). This dataset comprises proteins of 40 to 500 amino acids in length, for a minimum and a maximum of 282 and 4,173 heavy atoms. We also tested on the CASP14 and CASP15 datasets, to compare to the values reported by ESM-3. CASP14 and 15 structures were published after CATH4.2 and are thus not contained in 4.2. CASP14 and 15 contain proteins up to 2,265 residues in length with the biggest structure having 18,042 heavy atoms. To train the large bio2token model we leverage the Alphafold database (AFDB). We use a random sub-set of 100k clusters from FoldSeek's sequence-structure clusters (Barrio-Hernandez et al., 2023), and collect one structure per cluster.

**RNA:** We train on RNA3DB, which splits the RNA structures in the PDB into sequence-based and structural homology classes (Szikszai et al., 2024). The structures span a range of 2 to 4,450 nucleic acids in lengths with 42 to 95,518 heavy atoms. For training efficiency, we limit the training dataset to structures with maximum $10,000$ sequence length, but run inference on all lengths of the test set.

**Generalisation to complexes:** We test bio2token at inference time on multi-chain complexes and protein-RNA complexes. Note that neither multi-chain nor mixed complexes were included in the training.

## 3.2 Architecture and Training details

Figure 1B and C gives an overview of each layer composition and full architecture. Each layer of our encoder and decoder is a bi-directional implementation of the original *Mamba block*[1]. Between consecutive Mamba blocks we apply a layer norm.

### 3.2.1 Architecture studies

We ran various hyperparameter studies on a protein2token training with the CATH 4.2 protein dataset. We tested the effects of varying encoder and decoder layers on the model performances in terms of RMSE and found that, given limited compute, 4 encoder layers and 6 decoder layers to work best as a trade-off between model size and batch size. Additional details on the effect of the number of encoder layers is provided in Appendix A.2. We find the RMSE versus codebook size relationship to approximately follow a power law, see Fig. 6 in the Appendix A.2. Ultimately, the choice of codebook size will be a trade-off between accuracy and downstream modeling. A tokenizer with increasing vocabulary will make downstream generation models harder. We decided, for all of our models to stick with a codebook of 4096, which is in line with other published structure tokenizers, and allows for a fair comparison.

**Compressibility of tokens** To test the compressibility of the token sequences we train the tokenizer with an additional 1D convolutional layer before and after the quantizer network (pooling after the encoder and up-sampling before the decoder). We compress with $k \in [1, 2, 4]$, to shorten the all-atom sequence of length $N$ to $N/k$. Results are in Appendix table 4. RMSE increases by a factor of 1.7 and 2.6 for the compression factors of 2 and 4 respectively. This is similar to previously reported compressibilities for residue-level structure tokenizers (Gaujac et al., 2024)

**Computational efficiency: Mamba versus Invariant Point Attention** IPA is the most popular choice for structural modeling due to its SE(3) invariant properties. To compare Mamba versus IPA, we train a QAE with an encoder of 2 transformer layers, and a decoder with one IPA block with 4 recurrences (recycles). Although it is common in the literature to use 8 recurrences for the IPA block, it would restrict all-atom training to short proteins below approximately 100 residues given our GPU memory limit. With 4 recurrences we are able to fit one batch size of proteins of a maximum length of 2192 atoms (approximately 220 residues) on the GPU. With an equivalent Mamba-based architecture of 2 encoder and 4 decoder layers we can fit a batch size of 32 with maximum sequence length 2192. For a fair comparison we train both; a Mamba-based version with a batch size of 1 and a batch size of 32. Accuracies and run times are listed in Appendix A.2 table 5. We find that training protein2token with an IPA-decoder takes 3 times per step compared to the Mamba QAE. For a fixed compute budget of 24 hours the IPA-decoder reaches an RMSE of 2.2Å, compared to 0.8Å for the Mamba QAE with a batch size of 1, and 0.6Å for the Mamba QAE with batch size of 32. Although the Mamba-based QAE does not incorporate any rotational invariances in the architecture, it's learning efficiency is enough to make up for this lack of implicit bias. From a practical standpoint we conclude that an all-atom QAE tokenizer for large biomolecules is computationally intractable with standard IPA and conventional GPU hardware.

**Codebook efficiency: learned tokenizer versus spatial tesselation** To test if and by what factor the QAE approach to learning spatial vocabulary is indeed more efficient than assigning "spatial addresses" in a Voronoi tesselation (Voronoi, 1908; Aurenhammer & Klein, 2000) we compare our tokenizer errors to the error of a naive uniform voxelation of the space. The theoretical analysis is in the Appendix A.3. We find that a typical ribosomal RNA of spatial extent of 100Å requires 110k voxels to guarantee a 1Å accuracy. For a

---

[1]The Mamba block contains two branches; the selective SSM branch with a linear projection, followed by a one-dimensional convolutional layer and a nonlinear activation; and the skip connection branch that is a linear projection followed by a non-linear activation. This is directly imported from the implementation of (Gu & Dao, 2023)

desired RMSE of 0.2Å and a small molecule of spatial extent of 30Å tesselation of 191k voxels is necessary. This is more than a magnitude beyond the codebook sizes we employ.

### 3.2.2 Final model and training

We train four models, all with the same number of 4 encoder and 6 decoder layers, and a codebook size of 4096 for a total of 1.2M parameters (see section below for architecture study details). We use the Adam optimizer (Kingma, 2014), with polynomial learning rate scheduler and a starting learning rate of $3e^{-4}$. Depending on the model, we use 1 or 8 NVIDIA A10 GPUs (24GB / 184GB GPU RAM). We train three biomolecule specific models, *mol2token*, *protein2token*, and *rna2token*, respectivily trained on the $\nabla^2$DFT dataset, CATH4.2 dataset, and RNA3DB, and an harmonized *bio2token* model, trained on all three dataset and a subset of the AFDB dataset. Additionally, we use random rotation for data augmentation. Model specific parameters are:
**mol2token:** batch size=16, max seq length=64, 216k steps (44 hours), single GPU.
**protein2token:** batch size=16, max seq length=4160, 195k steps (68 hours), single GPU.
**rna2token:** effective batch size=32, max seq length=10000, 149k steps (38 hours),8 GPUs.
**bio2token:** effective batch size=32, max seq length= 10000, 257k steps (73 hours),8 GPUs.

We conduct an ablation study to evaluate the impact of additive architectural and training modifications on the performance of the Mamba QAE. All models are trained with identical quantization hyperparameters.

| Model + sequential modification | RMSE (CI ±95%) | Improvement (↓) |
|---|---|---|
| Mamba small [2 encoder / 4 decoder layers] | 0.72 ± 0.01 | - |
| + Data augmentation [Rotation] | 0.70 ± 0.01 | -1.91% |
| + Bi-directionality | 0.61 ± 0.01 | -12.89% |
| + Deeper [4 encoder / 6 decoder layers] | 0.55 ± 0.01 | -11.13% |
| + Inter-atomic distance loss | 0.52 ± 0.01 | -4.53% |

Table 1: Ablation study of final model and training choices. Ablation is run on protein2token training with the CATH 4.2 dataset.

We start with a baseline model consisting of 2 encoder and 4 decoder layers, and sequentially add data augmentation through random rotation, bi-directionality, deeper encoder and decoder, and finally the inter-atomic distance loss. The results are presented in Table 3.2.2. Modifying the original encoder/decoder layer to incorporate bi-directionality and increasing the number of layers resulted in a significant improvement, yielding a 22% reduction in reconstruction RMSE. Further enhancing the training strategy with random rotation augmentation and integrating an inter-atomic distance loss (as defined in Eq. 3), we observe a total RMSE reduction of 28% compared to the baseline.

### 3.3 Evaluation

To assess chemical or biochemical validity of the reconstructed structures, we benchmark our model on additional test metrics as outlined below.

**Small molecule validity test** We convert the heavy atom point clouds into molecules by inferring covalent bonds using atom type and inter-atomic distances with OpenBabel (O'Boyle et al., 2011). We first evaluate whether the recovered molecular system is equivalent to the encoded structure and then evaluate bond lengths, angles, and torsion angles using the methods described by Buttenschoen et al. (2024). We use RDKit to compute the energy of the conformer and compare to the average energy of 25 RDKit generated conformers (Landrum, 2013). We compute these statistics for test set ground-truth conformers and the reconstruction to evaluate any change. A reconstruction is said to pass all tests if it passes the tests from PoseBusters as well as produces the same molecular graph as the input.

**Proteins and RNA:** We report the template modeling score (TM-Score) between ground truth and reconstructed point clouds. It captures local and global structural alignment and

is designed to be size independent. The protein TM-score $TM_{prot}$ is calculated on the $C_\alpha$ of the amino acid back-bone (Zhang & Skolnick, 2004). The RNA TM-score $TM_{RNA}$ is calculated on the C3' of the nucleic acid back-bone (Gong et al., 2019). TM=0 means no structural similarity at all; TM=1.0 means structurally identical.

## 4 RESULTS

Table 2 summarizes the results of bio2token on all test sets. A more detailed version with separate analysis on back-bones and side-chains as well as the numeric results for the domain-specific tokenizers mol2token, protein2token, and rna2token, and their out-of-domain performance are provided in the Appendix tables 6, 7 and 8. Fig. 3 visualizes all reconstruction RMSEs on all biomolecular test sets with in-domain, out-of-domain and all-domain (bio2token) tokenizers.

| Best Model | Test-set | RMSE $\pm$ std (95% CI) [Å] | Validity Test |
|---|---|---|---|
| **Mol2Token on small molecules** | test-conformers
test-structure
test-scaffolds | **0.2±0.04 (0.01)**
**0.2± 0.04 (0.01)**
**0.2± 0.04 (0.01)** | 41.7% |
| **Bio2token on proteins** | CATH4.2 test
CASP14
CASP15 | **0.56±0.06 (0.01)**
**0.58±0.10 (0.02)**
**0.59± 0.11 (0.02)** | $TM_{prot}$: 0.98±0.01
$TM_{prot}$: 0.99±0.01
$TM_{prot}$: 0.98±0.02 |
| **Bio2token on RNA** | RNA3DB-test | **0.66± 0.21 (0.01)** | $TM_{RNA}$-score: 0.96 ± 0.12 |
| ESM-3 Tokenizer on proteins | CASP14
CASP15 | 1.3 ± 0.2
1.7 ±0.4 | – |
| InstaDeep on proteins | PDB sub-set | back-bone: 1.89 | $TM_{prot}$: 0.94 |

Table 2: Summary of the best tokenizer models: Atom-wise RMSE between the ground truth structure point cloud and the reconstructed point cloud from the tokens. Validity tests for small molecules are the chemical validity metrics as described in the main text and for proteins and RNA we provide the TM-scores as a measure of tertiary structural similarity.

**Small molecules:** mol2token reconstructs small molecule conformers of unseen molecules and unseen scaffold families with an average RMSE of 0.2Å versus 0.36Å for the combined model bio2token. Fig. 2A shows a valid reconstructed conformer. from the test set on top of the ground truth conformer. We found that 41.7% of all reconstructed molecules with mol2token passed all of our validity metrics and are similarly chemically valid as the ground-truth structures. We conclude that the mol2token vocabulary is sufficiently expressive to capture most molecular geometries in a manner that maintains their chemical structural properties in almost half the cases.

**Proteins:** bio2token outperforms protein2token on CASP14 and CASP15 test hold-outs with RMSE values around 0.58Å and 0.59Å versus 0.61Å and 0.8Å. This is significantly lower than ESM-3's decoder reconstruction on CASP14 (1.3Å) and 15 (1.7Å) that infers all-atom structure from the residue-level only encodings. InstaDeep's back-bone tokenizer compares with a back-bone RMSE of 1.89Å to bio2token's back-bone RMSEs of 0.52-0.55Å across the different protein test sets. Generally back-bone atoms have higher reconstruction accuracy than side-chain atoms for most test sets and tokenizers. protein2token was only trained on CATH4.2 proteins, which have sizes less than 500 residues and are smaller than several test set structures in CASP14 and 15, explaining why bio2token achieves much better results on these test sets. Generally the $TM_{prot}$ are all above 0.99, indicating that structural homology in terms of tertiary structure is highly preserved.

**RNAs:** bio2token reconstructs the RNA3DB test dataset with the lowest RMSE average of 0.66Å on all atoms, compared to 0.73Å for rna2token. Likely, bio2token is superior because it learned point cloud densities from magnitudes more data. The largest RNA chain in the RNA3DB test data is 8toc.R with 4,269 nucleic acids and a total of 90,441

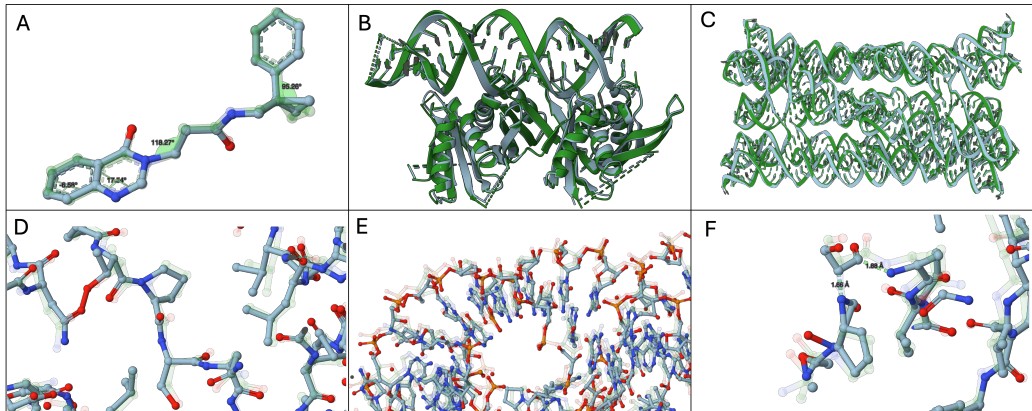

Figure 2: 3D renderings of ground truth molecules in green and reconstructions from decoded coordinates in blue. Ground truth molecules are made transparent in the ball and stick panels to make it easier to see the auto reconstructed models. Visuals prepared with Mol* (Sehnal et al., 2021) (A) Example from $\nabla^2$DFT scaffold split test set + mol2token reconstructed result. (B) RNA-Protein complex, PDB = 3WBM reconstruction by bio2token (C) Multi chain RNA complex, PDB = 7PTL. Reconstruction by bio2token (D) neighborhood of residue on loop of 3WBM found near center of coordinate space (E) close up of RNA helix of 3WBM (F) Example of errors found near edge of coordinate space.

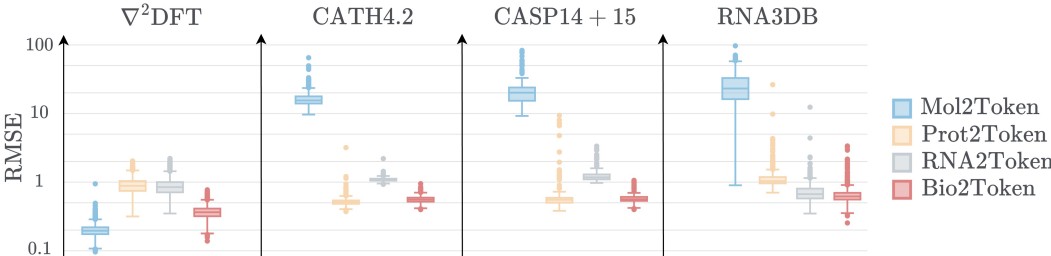

Figure 3: reconstruction results on all test data. Numeric values are provided in Appendix tables 6 - 8. For small molecules, only the domain-specific tokenizer mol2token and the combined bio2token achieve reasonable accuracy of 0.25-0.35 Å. For proteins (CATH4.2 test, CASP14/15) protein2token and bio2token achieve the best results. For the RNA3DB test set rna2token and bio2token have comparable results with reconstructions around 0.6Å. Macromolecules cannot be reconstructed from the small molecule mol2token vocabulary.

atoms. rna2token achieves a reconstruction RMSE of 1.53Å on this structure, compared to bio2token with 1.82Å.

**Complexes:** None of the tokenizers has ever seen a complex during training. Here, we tested to what degree we can encode and reconstruct RNA-protein and multi-chain complexes with the QAE. Fig. 2B shows a protein-RNA complex (pdb: 3wbm) with a combined residue count of 396 amino and nucleic acids with 3714 atoms, with an RMSE of 0.77Å. Figures 2 D,E, and F show close-ups of the complex. D and E are loop and helix regions with good reconstructions. F is a close up on the periphery of the coordinate space, where errors increase. Fig. 2C shows a multi-chain RNA complex (pdb:7ptl) reconstructed with bio2token at an average RMSE of 0.82Å. This complex comprises 720 nucleic acids with a total of 15,337 atoms.

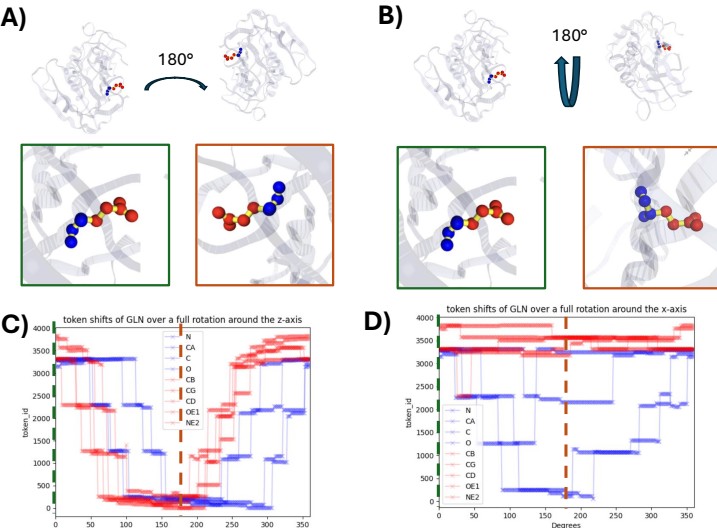

Figure 4: Token circularity with rotations. A and B visualise a $\pi$ rotation of the protein around the z- and x-axis. The zoom into the `GLN` amino acid shows how the individual atoms are changing orientations with respect to the centre. The respective token ids of each atom on the highlighted `GLN` are plotted in C) and D) as a function of rotation angle. The green and red dotted lines correspond to the tokens at the positions in A) and B).

### 4.1 INSIGHTS INTO WHAT BIO2TOKEN LEARNS

The tokenizer models learn to efficiently encode and decode point clouds of variable sizes, centered at the origin. Appendix A.5-Fig. 8 shows scatter plots for a sample of 10k points across all structure point clouds with their absolute distance to the centre and their RMSE. RMSE increases once the point's distance to centre increases past the common size range of the training structures. This can also be seen in Fig. 2F, where reconstructions deviate at the periphery of the coordinate space for a structure of about 16,000 atoms). Small molecules only span a few Angstroms and mol2token not surprisingly fails at reconstructing macromolecules (see Appendix table 8, a mol2token reconstructed protein looks like a dense point potato, not shown). protein2token and rna2token can tokenize RNAs and proteins as they share similar point cloud sizes and densities. This motivates the combined tokenizer training of bio2token, which performs best for proteins, RNA and protein-RNA complexes as it has seen the most diverse point configurations during training.

**Rotational Variance**   bio2token does not exploit rotational invariance in it's architecture. Here, we show that sub-Angstrom accuracies are achievable without those inductive biases. The bio2token tokens are varying periodically with respect to rotations. To visualise the effect we show the individual amino acid `GLN`  and its back-bone and side-chain atoms under a set of full $2\pi$ rotations around the z- and the x-axis. Fig. 4 shows how the atom token ids shift with respect to changes in orientation. Reconstruction errors are not biased towards any orientation, see Appendix A.5.2 Fig. 9.

## 5 DISCUSSION

In this study, we explored the potential of Mamba to encode high-resolution structures for diverse biomolecules. Specifically, opting for a simple Mamba-based architecture, instead of rotationally invariant IPA, allows us to scale trainings to large biomolecules that cannot be trained on with the latter approach with conventional GPU hardware. In fact, we show that reconstruction accuracies around $0.5 - 0.6$Å  from a vocabulary of 4096 is achievable without any SE(3) invariance in either features, or network architecture. Our combined tokenizer, bio2token, demonstrates that encodings can be learned across different classes of

macromolecules once encoded on the common atomistic resolution. The comparable small amount of data (127,000 macromolecules in total) used in our trainings signals that all-atom encoding might substantially enhance training efficiency, compared to more coarse-grained encoding that lack atomistic detail, and leverages more information from the structures sampled. Atomistic detail is important for many biomolecular design applications – the precise positioning of individual atoms within a protein or RNA molecule can significantly impact its function and interactions with other molecules. Furthermore, the reliance on separate models for different components of a molecule, (e.g., back-bones versus side-chains), can introduce additional complexity and potential sources of error. Our work highlights the potential of leveraging models like bio2token to improve the design and modeling of biomolecules and biomolecular complexes.

### 5.1 Limitations and future directions

Although our model achieves low RMSE values, having low RMSE is not necessarily sufficient to guarantee that the reconstructed molecule is chemically valid. Even small deviations in decoded atom coordinates can result in structures with steric clashes, improper covalent bonds, or improperly strained geometries. For example, as seen in Fig. 2F, our reconstructed structures can deviate from valid molecular geometries, for example implying covalent bonds where there should not be and vice-versa missing bonds where there should be. One potential solution could be to train the model on additional data which may further lower RMSEs to a level that produces more chemically valid biomolecules without having to hard code constraints into the losses of our models. Alternatively, we could employ heuristic and physics based post-processing protocols like those used in Abramson et al. to help translate generated point clouds into valid molecules.

Nonetheless, our work points towards Mamba-based architectures as a viable and promising alternative to transformer-based methods for modeling atomic-resolution biomolecular structures. In the quantized QAE form as presented here, our models could facilitate compatibility with language models and we leave the joint representation of atom identity and coordinates to future work. The continuous embedding (no quantizer), combined with compression, could provide useful latent spaces for methods like flow matching or diffusion. As such, we anticipate that our work could connect to many potential downstream modeling applications in chemistry and biology.

## 6 Code Availability

Code and model weights are available for all trained tokenizers. Inference scripts are provided for pdb formated files at `https://anonymous.4open.science/r/bio2token-72F2`

## 7 Acknowledgements

To be filled after review.

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

# A   APPENDIX

## A.1   DATASETS

| Dataset | Dataset size and splits | Structure size | Used in |
|---|---|---|---|
| $\nabla^2$DFT | **train**: 8.9M conformers (0.5M molecules) 
 **test-conformer**: 1.5M conformers (1.5M molecules) 
 **test-structure**: 1.2M conformers (176k molecules) 
 **test-scaffold**: 1.1M conformers (177 molecules) | atoms min: 8 
 atoms max: 27 | Mol2Token, Bio2Token |
| CATH4.2 

 CASP14 

 CASP15 | **train**: 17k structures 
 **test + val**: 1.6k structures 
 **test**: 88 structures 

 **test**: 155 structures | res/atoms min: 40/282 
 res/atoms max: 500/4.2k 
 res/atoms min: 49/401 
 res/atoms max: 2.2k/18k 
 res/atoms min: 46/341 
 res/atoms max: 10k/7.9k | Protein2Token, Bio2Token |
| RNA3DB | **train**: 10k structures 
 **test**: 1.4k structures | res/atoms min: 2/42 
 res/atoms max: 4.5k/96k | RNA2Token, Bio2Token |
| AFDB sample | **train**: 100k structures | res/atoms min: 21/174 
 res/atoms max: 2.7k/22k | Bio2Token |

Table 3: Summary of training and test datasets, including minimum and maximum number of residues and atoms.

## A.2   ARCHITECTURE STUDIES

**Effect of number of encoder blocks**   The encoder mixes the atom coordinates and the degree of mixing, or "spread" across atom positions is determined by the number of encoder Mamba blocks and hidden state size. To quantify the spread of local information we define the mixing radius as the number of positions that change their token id when the atom at position $i$ is deleted. Here, we fix the hidden state size of 128 and train QAEs with increasing numbers of encoder blocks $n_{enc} = [2, 4, 5, 6]$ and find the mixing radius to be almost linear with a best fit for a second order polynomial, see Figure 5. This relationship is similar to what is expected from a convolution. For example 2 blocks result in a mixing of $\pm 2.7$ positions to the left and right; and 6 blocks mix $\pm 5.3$ positions.

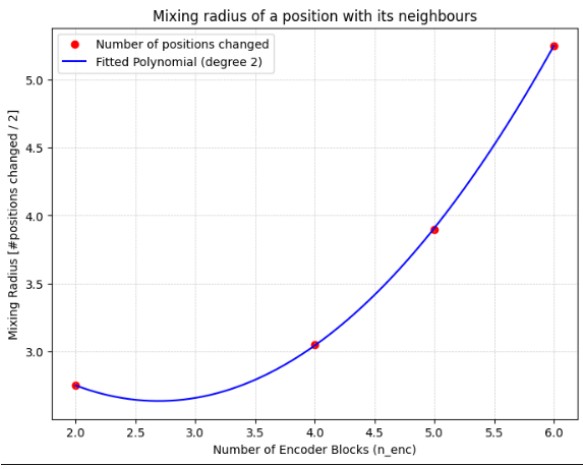

Figure 5: Average mixing radius of per-atom position information with increasing number of Mamba blocks in the encoder.

**Codebook size**   We train protein2token on the CATH4.2 dataset, with a fixed model size. We vary codebook sizes by increasing quantization dimensions $D \in [4, 5, 6, 7, 8]$ with a

fixed level of $L = 4$, for total codebook sizes of $[256, 1024, 4096, 16348, 65536]$. We find the accuracy versus codebook size relationship to approximately follow a power law, see Fig. 6. Ultimately, the choice of codebook size will be a trade-off between accuracy and downstream modeling. A tokenizer with increasing vocabulary will make downstream LLM generation harder. For the final training of bio2token we chose 4096 as our codebook size, which is in line with other published structure tokenizers, and allows for a fair comparison.

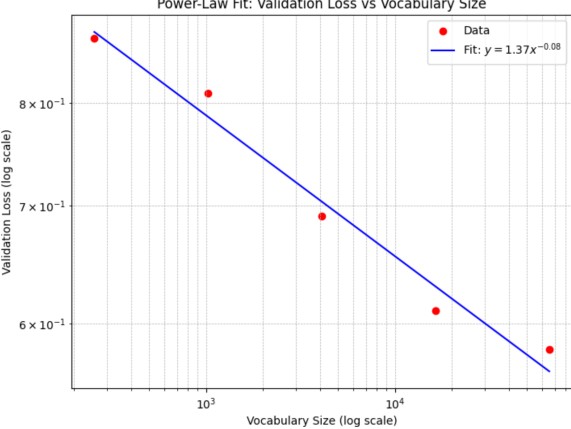

Figure 6: Protein2token (CATH dataset) reconstruction accuracy as a function of codebook size.

**Compressibility of tokens** We train protein2token (CATH dataset) with compression factors of $[1, 2, 4]$ using 1D convolutional layers. Table 4 below shows the the relationship between test set reconstruction RMSD and compressibility factor with a codebook size of 4096. We also tested if increasing the SSM's hidden size could increase compressibility, which we found to not be the case.

| Compression | D_model hidden size | RMSE [Å] | factor of RMSE increase |
|:---:|:---:|:---:|:---:|
| 1 | 128 | 0.86 | — |
| 2 | 128 | 1.49 | 1.7 |
| 4 | 128 | 2.22 | 2.6 |
| 1 | 1280 | 0.84 | — |
| 2 | 1280 | 1.45 | 1.7 |
| 4 | 1280 | 2.15 | 2.6 |

Table 4: Effect of compression on accuracy RMSE. Interestingly, increasing the hidden state size does not help noticeably to recover accuracy.

A.3 Model efficiency comparisons

**Computational efficiency and performance: Mamba versus IPA** We train a protein2token tokenizer with a 2-layer transformer encoder and an IPA decoder with 4 recyclings. Due to GPU memory constraints, training is limited to protein structures of a maximum length of 2192 atoms, at a batch size of 1. We train an equivalent Mamba-based protein2token with 2 encoder Mamba-blocks and 4 decoder Mamba-blocks, with a batch size of 1 and the maximum batch size before GPU memory is exhausted, which is 32. We find that the IPA-based QAE requires 1 sec/step, compared to 0.3sec/step for an equivalent Mamba-based QAE. In terms of achieved validation accuracy IPA-based architecture is significantly worse than the Mamba-based QAE with an RMSE of 2.18 versus 0.81. Likely this is due to the "small" number of IPA-block recycles, often 8 (instead of 4) are cited in the literature. But this becomes prohibitive for sequences lengths of 2192. To compare at the

full capacity of the GPU hardware, we find that training for 24 hours with the Mamba-based QAE with a maximum batch size of 32 has superior accuracy with 0.62Å.

| Architecture | Time [sec/step] | Validation accuracy after 24h run time [Å] | Validation accuracy after 70k steps [Å] |
|---|---|---|---|
| Transformer encoder, IPA decoder, batch size = 1 | 1.0 | 2.18 | 2.18 |
| Mamba, batch size = 1 | 0.3 | 0.81 | 0.91 |
| Mamba, batch size = 32 | 0.7 | 0.62 | 0.65 |

Table 5: Effect of compression on accuracy RMSD. Increasing the hidden state size does not recover accuracy.

**Codebook efficiency: learned versus spatial tesselation**  We explore how well the trained QAEs perform relative to idealized voxel partitions and learned voronoi tesselations. For a desired tesselation resolution $a$ (the side length of a voxel), and a total cubic volume of side length $A$ (the maximum spatial extent of biomolecular structures) results in a total number of voxels $N_v = (A/a)^3$. To calculate the average reconstruction accuracy of a point (atom) in a voxel, we calculate the average $rmsd_v$ to the voxel centre:

$$rmsd_v = \frac{8}{a^3} \int_0^{a/2} \int_0^{a/2} \int_0^{a/2} \sqrt{x^2 + y^2 + z^2}\, dx\, dy\, dz$$

With Monte-Carlo integration (not shown) this is approximately $0.48 \times a$. To tesselate a biomolecular structure of cubic volume with a side length $A$ and a desired average accuracy $rmsd_v$, a total voxel count of

$$N_v = \left( \frac{0.48 \times A}{rmsd_v} \right)^3$$

Figure 7 plots the number of total Voronoi voxels needed to encode the 3D space of three exemplar cubes of side length $a = [10, 60, 80]$Å , representative for small molecules, proteins and RNA respectively . We center structures at zero, sample rotations and use k-means clustering to find 4096 cluster centers that are used as the centroids of Voronoi tesselations. Upon comparing these approaches we see that for the tested codebook size, the QAE approach achieves lower $rmsd_v$ , suggesting that it learns beyond the atom coordinate aaddress.

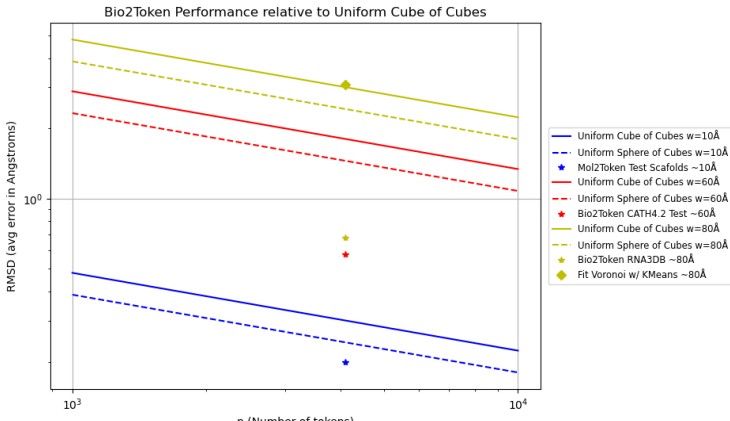

Figure 7: Comparing the reconstruction error between learned tokenizers, trained with 4096 codebook size, a naive tesselation of increasing number of voxels and a k-means Voronoi tesselation approach.

## A.4 TOKENIZER RESULTS

| Model | Test-set | RMSE $\pm$ std (95% CI) [Å] | Validity Test |
|---|---|---|---|
| **Bio2Token on small molecules** | test-conformers
test-structure
test-scaffolds | **0.36±0.07 (0)**
**0.37± 0.07 (0)**
**0.36± 0.07 (0)** | $< 1\%$ |
| **Bio2Token on proteins** | CATH4.2 test | bb: 0.52± 0.07 (0.01)
sc: 0.59± 0.06 (0.01)
**all: 0.56±0.06 (0.01)** | TM$_\text{prot}$: 0.98±0.01 |
| | CASP14 | bb: 0.54±0.10 (0.02)
sc: 0.62±0.09 (0.02)
**all: 0.58±0.10 (0.02)** | TM$_\text{prot}$: 0.99±0.01 |
| | CASP15 | bb:0.55±0.12 (0.02)
sc:0.63± 0.12 (0.02)
**all: 0.59± 0.11 (0.02)** | TM$_\text{prot}$: 0.98±0.02 |
| **Bio2Token on RNA** | RNA3DB-test | bb: 0.66± 0.21 (0.01)
sc: 0.65 ± 0.22 (0.01)
**all: 0.66± 0.21 (0.01)** | TM$_\text{RNA}$-score: 0.88 ± 0.12 |
| ESM-3 tokenizer on proteins | CASP14 | back-bone:0.61 ± 0.1
**all: 1.3 ± 0.2** | |
| | CASP15 | back-bone: 1.0 ± 0.3
**all: 1.7 ±0.4** | |
| InstaDeep tokenizer on proteins | self-defined test set from the PDB | back-bone: 1.89
side-chains not modeled | TM$_\text{prot}$: 0.94 |

Table 6: Bio2token results: Atom-wise RMSE between the ground truth structure point cloud and the reconstructed point cloud from the tokens. "bb" and "sc" are the respective RMSEs over the back-bone and side-chain atoms in the case of proteins and RNAs. Bio2token is unable to preserve chemical validity of small molecules and mol2token should be used for these structures. For proteins and RNA we provide the TM-scores as a measure of tertiary structural similarity.

### A.4.1 IN-DOMAIN TOKENIZING

| In-domain tokenizing | Test-set | rmse ± std, (95% CI) [Å] | Validity Test |
|---|---|---|---|
| **mol2token on small molecules** | test-conformers
test-structure
test-scaffolds | **0.20± 0.04(0.01)**
**0.20± 0.04 (0.01)**
**0.20± 0.04 (0.01)** | 41.7% passed all chemical validity metrics |
| **protein2token on proteins** | CATH4.2 test | bb: 0.49±0.12 (0.01)
sc:0.56±0.11 (0.01)
**all: 0.53±0.12 (0.01)** | $TM_{prot}$: 0.99±0.01 |
| | CASP14 | bb: 0.57±0.21 (0.04)
sc: 0.65±0.21 (0.04)
**all: 0.61±0.21(0.04)** | $TM_{prot}$: 0.99±0.01 |
| | CASP15 | bb:0.76±1.21 (0.19)
sc:0.85±1.25 (0.20)
**all: 0.80±1.23 (0.19)** | $TM_{prot}$: 0.99±0.03 |
| **RNA2token on RNAs** | RNA3DB-test | bb: 0.73±0.34 (0.02)
sc: 0.72 ±0.40 (0.02)
**all: 0.73±0.39 (0.02)** | $TM_{RNA}$-score: 0.86 ± 0.13 |
| ESM-3 Tokenizer on proteins | CASP14 | back-bone:0.61 ± 0.1
**all: 1.3 ± 0.2** | |
| | CASP15 | back-bone: 1.3 ± 0.3
**all: 1.7 ±0.4** | |
| InstaDeep | self-defined test set from the PDB | back-bone: 1.89
side-chains not modeled | $TM_{prot}$: 0.94 |

Table 7: In-domain tokenizing: The reconstruction error is the atom-wise rmse between the ground truth structure point cloud and the reconstructed point cloud from the tokens. "bb" and "sc" are the respective rmses over the back-bone and side-chain atoms in the case of proteins and RNAs. Validity tests for small molecules are the chemical validity metrics as described in the main text and for proteins and RNA we provide the TM-scores as a measure of tertiary structural similarity

### A.4.2 OUT-OF-DOMAIN TOKENIZING

| Out-of-domain tokenizing | Test-set | rmse ± std (95% CI) [Å] | Validity Test |
|---|---|---|---|
| mol2token on proteins | CATH4.2 test
CASP14
CASP15 | all: 16.40± 4.07 (0.24)
all: 21.37± 10.44 (2.18)
all: 23.23± 13.95 (2.20) | $TM_{prot}$: 0.13±0.04
$TM_{prot}$: 0.13±0.05
$TM_{prot}$: 0.13±0.06 |
| mol2token on RNA | RNA3DB-test | all: 25.88±12.22 (0.65) | $TM_{RNA}$: 0.02 ±0.01 |
| protein2token on RNAs | RNA3DB-test | all: 1.16±0.79 (0.04) | $TM_{RNA}$: 0.81 ±0.16 |
| RNA2token on proteins | CATH4.2 test
CASP14
CASP15 | all: 1.09±0.07 (0.01)
all: 1.27±0.36 (0.08)
all: 1.30±0.39 (0.06) | $TM_{prot}$: 0.96±0.03
$TM_{prot}$: 0.96±0.04
$TM_{prot}$: 0.96±0.04 |

Table 8: Applying tokenizers on out-of-domain molecules. Only all-atom rmses are shown here for simplicity. mol2token to proteins and RNAs: The rmse values show the insufficiency of learning larger biomolecular structures just from small molecules. protein2token on RNAs: The rmse if higher than the rna2token reconstruction error (reported in the main text), but is in close proximity. rna2token on proteins: the rmse is slightly worse than the protein2token errors reported in the main text on CATH4.2 and CASP14, but better on CASP15.

## A.5 INSIGHTS INTO BIO2TOKEN

### A.5.1 RMSE PER ATOM AS A FUNCTION OF DISTANCE TO CENTRE

Figure 8 shows scatter plots for a sample of 10k points across all structure point clouds with their absolute distance to the centre and their RMSE

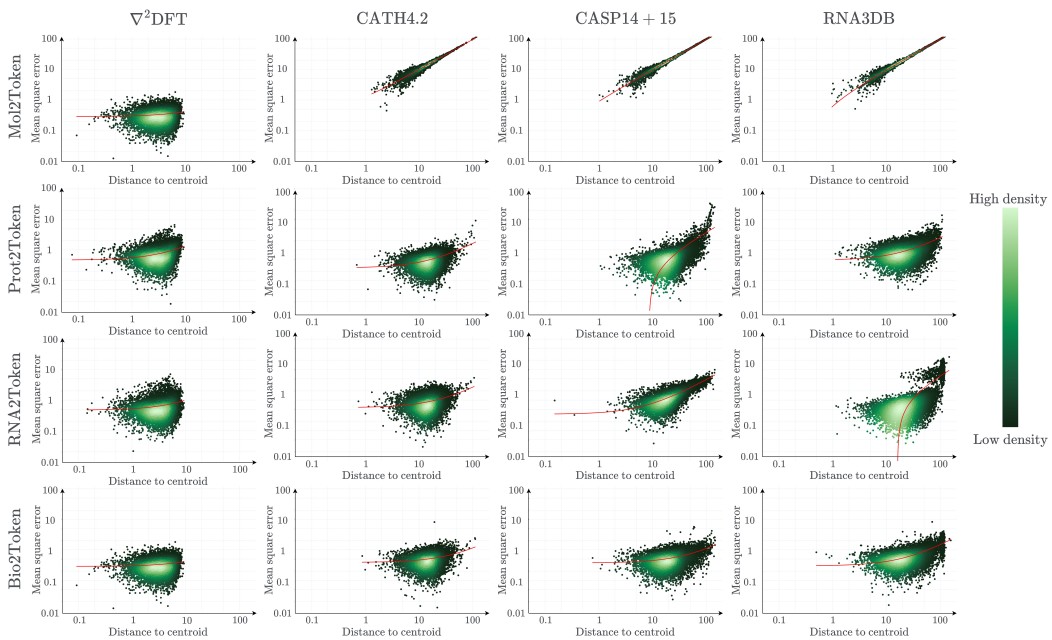

Figure 8: Reconstruction RMSE per point as a function of its distance to the centre. Each subplot is a random sample of 10.000 points across all point clouds of the respective dataset.

### A.5.2 RECONSTRUCTION ERRORS ARE INDEPENDENT OF ROTATIONS OF THE STRUCTURE

Figure 9 shows the RMSD for an exemplar proteins, rotated around the x-, y- and z-axes.

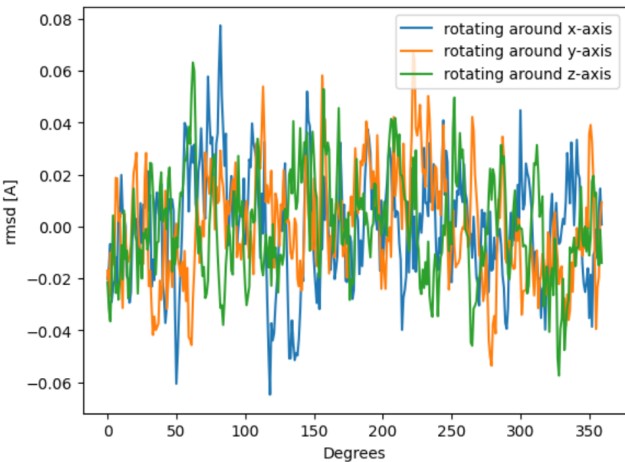

Figure 9: The reconstruction error of an exemplar protein under a full set of $2\pi$ rotations around all major axes. The reconstruction error shows no orientation bias.

