# OpenReview forum: "bio2token: all-atom tokenization of any biomolecular structure with mamba"
_ICLR.cc/2025/Conference — Submitted to ICLR 2025_

### Official Review · Reviewer_odDL · 2024-10-29

**Soundness:** 3
**Presentation:** 3
**Contribution:** 3
**Rating:** 8
**Confidence:** 3

**Summary:**

The paper describes mamba-based approach to representation and reconstruction of atomic configurations of small molecules, proteins, and RNA. The paper reports ability of the presented appoach to scale to 10^5 atoms while reaching competitive accuracies.

**Strengths:**

The paper describes a meaningful effort in modeling mechanistically-motivated molecular structure, such as Cartesian coordinates of the atoms in covalent systems, instead of various serialized descriptors. This is a strong point towards originality and significance.

The paper reports application of the model to biochemically relevant molecular systems, including small molecules, proteins, and RNA, described in publicly available datasets. Ability to treat configurations of covalent systems up to 10^5 atoms is significant.

Adaptation of mamba to the problem enables efficiency of all-atom modeling, including small size of the model, fast inference, and attractive scaling with molecular size. The referee is aware of several ongoing mamba-based developments for computational chemistry, the reported one is definitely original and useful.

**Weaknesses:**

While paper explicitly claims ability to "reach competitive accuracies" there are no mentions of accuracies to compare with. In other words, the model performance does not seem to be evaluated against any alternatives.

Atomic configurations of molecules are a staple of computational chemistry. There is a literature on AI modeling of Cartesian coordinates of molecules in computational chemistry. It would be fair for the authors to cite such contributions, even those limited to small molecules.

There's a statement and evidence of scaling to large system size, but no scaling curves reported.

**Questions:**

What is the nature of scaling with the molecule size? Is improved accessibility of large systems a consequence of improved scaling or improved prefactor?

What are existing approaches to representation and reconstruction of atomic configurations in computational chemistry?

What are performance metrics related to the chemical correctness of the reconstructed configurations? It is straightforward to create bonding patterns based on interatomic distances of reconstructed configurations and compare them with the ground truth patterns.

---

> ### Author Response · Authors · 2024-11-18
>
> Thank you for reviewing and providing constructive comments! As we prepare the requested analysis – could you please clarify W1&Q1 for us?
>
> ### W1&Q1: Lack of performance comparison
>  “[...] there are no mentions of accuracies to compare with. [...] the model performance does not seem to be evaluated against any alternatives.” -- Is this question with respect to the existing AI methods for atomic coordinate reconstruction (W2&Q2)? Because we do provide comparisons on the auto-reconstruction errors on the same test hold outs as ESM-3's and InstaDeep’s tokenizer models. These models are optimized on the same objective of auto-reconstruction, which makes it “apples-to-apples". We provided those results in tables 2 and 3 in the appendix and mention it in the main text in the results section – lines 300 onwards. We can make the wording more clear and/or bring numerical tables into the main text if helpful.
>
> ### W3&Q1: Lack of scaling analysis
> The referee rightfully points at the lack of a numerical comparison on computational efficiency and scaling. We are currently running computational efficiency experiments, swapping Mamba layers for IPA transformers to compare run times and length scaling. We will report back over the next week.
>
> ### W2&Q2: Missing citations and reference to previous work
> Very fair, we missed citing several relevant ML methods for cartesian coordinate modeling in chemistry and will add relevant pre-work (e.g. https://arxiv.org/abs/2305.05708) . We will notify the referee once the amendments are ready for review.

---

> ### Author Response · Authors · 2024-11-25
> **Revised manuscript**
>
> We have revised our manuscript, altered sections are highlighted in red (we will remove the red before the end of the rebuttal period).
> We provide results to several of the reviewers requests.
>
> ### W1: The model performance does not seem to be evaluated against any alternatives
> Incorporated. We now provide a comparison with an IPA-based decoder (that is SE(3) invariant), which is the most common approach to structure modeling and is used in several structure decoders. We provide a training with an IPA-decoder on proteins, training with a batch size of 1 with a maximum length of 2k atoms (approx. 220 residues), which is the maximum limit for our GPU. We train the equivalent Mamba-based tokenizer with the same batch size of 1 and find the IPA step to take three times as long. After 24 hours of training, we find the Mama-QAE to have an accuracy of 0.8A, versus 2.2A for the IPA implementation. In practice, we can afford training the Mamba QAE with a batch size of 32 on the GPU, which further improves the performance. In summary, the Mamba QAE runs faster, and although not incorporating SE(3) invariance, learns efficiently. The results are in the main text section 3.2.1, lines 236 onwards, and the efficiency table is in Appendix A.3. Table 5.
>
> ### Q1: What is the nature of scaling with the molecule size
> See W1
>
> ### W2/Q2: There is a literature on AI modeling of Cartesian coordinates of molecules
> Incorporated. We now provide a dedicated paragraph in the introduction "3D structure tokenization" that summarizes relevant previous work in the field of 3D structure vocabulary learning, including proteins and small molecules.
>
> ### Q3: What are metrics related to the chemical correctness of the reconstructed configurations? It is straightforward to create bonding patterns based on interatomic distances of reconstructed configurations and compare them with the ground truth patterns.
> For the small molecules we use the protocol of chemical validity metrics as provide by the PoseBusters paper (Buttenchoen et al), see main text, section 3.3.
> To calculate bond lengths, torsion angles etc. for reconstructed proteins and RNA is indeed straightforward. But to be totally honest here: We forgot this comment and didn't work on this over the last week. Too many reviewers... apologies :(

---

### Official Review · Reviewer_6xss · 2024-11-03

**Soundness:** 2
**Presentation:** 2
**Contribution:** 3
**Rating:** 5
**Confidence:** 5

**Summary:**

It provides an all-atom level VQVAE for different modalities with Mamba, but the writing of this paper should be improved and additional experiments are needed for the evaluation.

**Strengths:**

- It provides an all-atom level VQVAE for different modalities, i.e., protein, RNA and ligands, which is the first work in this field.
- With mamba architecture, it is more efficient than transformer-based methods and maintains similar to superior performance.

**Weaknesses:**

- It is more like a technical report than a research paper. Modeling biomolecules in an all-atom resolution typically requires many complex operations such as the **broadcasting** (token index to atom) and **aggregation** (atom index to token) in AlphaFold3. Additionally, the model architectures will become complex either, such as **AtomAttentionEnconder** and **AtomAttentionDeconder** in AlphaFold3. However, this paper lacks details on these components and instead frequently mentions “Mamba” without substantive explanation.
- For evaluation of structure reconstruction, **pLDDT** (or, preferably, **pAE**) should be set as output heads .
- I question whether a codebook size of 4096 is sufficient to capture an all-atom vocabulary effectively. A comparison across different codebook sizes should be included.
- The low-quality reconstruction samples should also be visualized to help learn the issues of tokenizer.
- For complex structure reconstruction, multi-chain permutation alignment (as in AF2-Multimer) is usually necessary. However, this paper does not include details on that.
- It is impressive that the model achieves comparable results to CASP14/15 benchmarks with ESM3, despite being trained on only 18k CATH 4.2 dataset entries, as opposed to larger datasets like PDB, AFDB, or ESMAtlas used in ESM3. **Additional training details must be provided** to clarify how this performance was achieved.

**Questions:**

- **Efficiency IPA Transformer versus Mamba** section is overly brief. No tables or figures are provided.
- All-domain tokenizing Table could be moved to the main text.
- No code is available.

---

> ### Author Response · Authors · 2024-11-18
>
> Thank you for the review. We would like to clarify several comments and also ask for further clarifications to fully address the referee's concerns.
>
> ### Q3: There is no code.
> The code should have been accessible via the link in the section “Code availability”, if you scroll all the way down to the end of the manuscript. Please let us know if the link does not work. We post it here again: https://anonymous.4open.science/r/bio2token-72F2
> ### Q1: What is the performance and efficiency comparison between Mamba and IPA
> Very good call out, we are currently running the experiment on small molecules (all-atom on proteins is computationally prohibitive with IPA). We will report back once results are ready and manuscript is updated.
>
> ### Q3: Please move appendix table to main manscuript
> It is an important result, and we are happy to move it.
>
> ### W1: “Modeling biomolecules in an all-atom resolution typically requires many complex operations [...] However, this paper lacks details on these components and instead frequently mentions “Mamba”  "
> The lack of detail is real – please take a look at the code and try inference on the github link. Our paper shows that for the case of structure tokenization, no computationally expensive attention mechanisms like IPA or geometric attention are needed to achieve comparable performance (with cheaper compute :) ). This goes back to the referee's request for a proper performance and efficiency comparison. The experiment is currently running, we will report results and update the manuscript once finished.
>
> ### W2: “For evaluation of structure reconstruction, pLDDT (or, preferably, pAE) should be set as output heads.”
> This paper does not present de-novo generation. But the reviewer is indeed correct, that for down-stream generation a separate pLDDT /pAE head is necessary.
>
>
> ### W3: Effect of codebook size
> Fair request and we should add those details. We will update the manuscript and report back over the next week.
>
> ### W4: “Additional training details must be provided”.
> Could you be more specific what is missing? In section 3  “Experimental Details” it lists: number of layers, quantization levels, model size, optimizer, learning rates, batch sizes, GPU model and specifications, number of steps and total training time. We also provide a data table with number of samples and min and max number of atoms/sample. We did notice we forgot to list the hidden dimension sizes of the Mamba layers, which is indeed an important factor.
>
> ### W5: Multi-chain permutation alignment (as in AF2-Multimer) is usually necessary”
> This method does not distinguish between a multi-chain or a single chain complex. They both are single structural point clouds. No alignments are needed as complexes are treated as a whole.

---

> ### Author Response · Authors · 2024-11-25
> **Updated manuscript incorporating requested changes**
>
> We uploaded a revised version of the manuscript, incorporating the reviewers critique. We highlighted altered sections in red (we will change back to black before the end of the rebuttal period).
> ### W1a: Modeling biomolecules [...] typically requires many complex operations such as the broadcasting (token index to atom) and aggregation (atom index to token) [...]
> These mixing operations are achieved by the Mamba encoder and decoder blocks, which can in a simplified manner be understood to act like a convolution. To provide more insight how the number of encoder blocks influences the "token mixing radius" (or as the reviewer describes it as broadcasting) -- the number of atom positions that influence the token at a given atom position, we now provide a dedicated study in Appendix A.2., Fig 5. The more encoder layers are chosen, the more mixing occurs across positions in an approximate linear relationship.
>
> ### W1b: The paper lacks details on architecture components
> We provide extensively more detail on the architecture and hyperparameter choices in the revised version. A detailed overview of each encoder/decoder layer, including Mamba blocks, normalization layers and bi-directionality are added to Figure 1 and a thorough description is provided in a dedicated section "3.2 Architecture and Training Details". To provide more insight into which aspects of the architecture and which parameter choices are most important for performance, we further provide an ablation study on the final model, where we show that bi-directionality and model size are most important (table 1 in the main text).
>
> ### W3a: A comparison across different codebook sizes should be presented
> Incorporated. We now provide a study of codebook sizes from [256...65,000] versus RMSE, please see Appendix A.2 Figure 6. We show that RMSE can be decreased with increasing codebook size according to a power law. So with sufficient model sizes bigger vocabularies can be learned for better RMSE, but that defeats the purpose of the quantization compression for downstream generation.
>
> ### W3b: I doubt a codebook size of 4096 is sufficient for all-atom structures.
> Our RMSE are all well below one Angstroms on all test sets. Our small molecule reconstructions are around 0.2 Angstroms and are chemically valid in 42% of the cases according to PoseBusters validity criteria. What would the reviewer regard as sufficient?
> ### W4: The low-quality reconstruction samples should also be visualized to help learn the issues of tokenizer
> Incorporated. Please see Figure 2F for an example of a poor reconstruction at the periphery of the coordinate space.
> ### W5: More training details must be provided.
> Incorporated. See answer to W1b.
>
> ### Q1: Efficiency comparison of Mamba versus IPA
> Incorporated. We now report a computational and performance comparison of the Mamba-based QAE versus an IPA decoder implementation. See section 3.2.1 paragraph "computational efficiency: Mamba versus IPA", as well as Appendix A.3. We find that an IPA decoder runs three times slower per step and can only be trained with a batch size of 1 when scaling to proteins of approximately 220 residues with about 2000 atoms. The equivalent Mamba QAE can be run with a batch size of 32. Even when trained with a batch size of 1 we find that Mamba QAE outperforms the IPA implementation.
>
> ### Q2: Move table
> Done.
>
> ### Q3: No code
> The code link is at the bottom.

---

> > ### Author Response · Authors · 2024-11-30
> >
> > We hope we fully addressed their concerns by the additional material incorporated in the revision in response to their feedback. We'd appreciate their feedback on the revision and are looking forward to their comments.

---

### Official Review · Reviewer_rmoF · 2024-11-03

**Soundness:** 3
**Presentation:** 2
**Contribution:** 2
**Rating:** 5
**Confidence:** 4

**Summary:**

This paper presents a novel approach for efficient encoding and representation of large 3D molecular structures at the all-atom level. The authors develop quantized auto-encoders that learn atom-level tokenizations of complete proteins, RNA, and small molecule structures with high reconstruction accuracy. The Mamba state space model architecture employed is shown to be computationally efficient, requiring less training data, parameters, and compute compared to transformer-based methods, while maintaining similar or superior performance. The authors demonstrate the ability to scale to biomolecular systems with up to 95,000 atoms, which is beyond the capabilities of existing transformer-based models. The learned structure tokens from this approach, called bio2token, may serve as the input for future all-atom language models.

**Strengths:**

- For the first time, Mamba is used to construct an all-atom discrete representation of multiple biological structures.
- The generalization capability of bio2token for complexes is also quite impressive.

**Weaknesses:**

- The definition of biological structures in the article only involves coordinate point clouds, which is incomplete; information on atomic types is also crucial.
- The paper only discusses discrete tokenization without demonstrating the advantages of this tokenization through downstream applications. Moreover, bio2token does not reduce the number of atoms, raising doubts about whether it can truly support language model development in the relevant field.

**Questions:**

- The excellent generalization ability of bio2token for complexes might simply be due to its replication of the input coordinates.
- There are many other types of compounds that were not covered in this paper, and furthermore, no corresponding quantitative analysis was included.
- There is also no quantitative analysis in the discussion regarding computational efficiency.

---

> ### Author Response · Authors · 2024-11-18
>
> Thank you for the relevant critique.
> To address all concerns, may we ask for some clarifications.
> ### Q1: Is the excellent generalizability due to its replication of the input?
>  Using an auto-encoder model to learn tokens is a common approach and mimics the approach taken in ESM-3 and InstaDeep’s structure tokenizer (see manuscript "Related Work"). The loss is the auto-reconstruction loss (“replication of input”). Having “excellent generalization” speaks for the tokens/vocabulary to well capture the spatial conformations of the structures and is a desired property. Would the referee prefer more analysis on the token interpretability and what they encode? How do we best clarify your comment and what would be most helpful?
> ### Q2: “Compounds and more quantitative analysis is missing”
> Could you be specific on the type of compounds besides the small molecules, proteins and RNA? Something exotic -- e.g a protein-glycan complex? What analysis would you regard as convincing?
> ### Q3: Computational efficiency comparison is missing
> This critique is spot-on. Given that the architecture is a key differentiating factor this analysis is missing. We will follow up on this over the coming week with a thorough numerical comparison.
>
> ### W1: No atomic identity information is incorporated
> We assume the referee is forward-thinking to generative tasks. In the case for downstream generation, atom/residue idenity inforamtion is required and can be handled separately (e.g. ESM-3 uses independent tracks for structure and residues). We would like to stress that this is a **structure** encoder, hence we didn't opt for a joint sequence-structure encoding, which complies with all common SOTA structure tokenizers (InstaDeep, ESM-3 tokenizer etc.). Does this clarify your comment? Did you find the manuscript unclear w.r.t. rational?
>
> ### W2: No compressibility or downstream generation is demonstrated.
> This is a fair weakness. We are currently running compression experiments and will report the results over this week as they solidify. However, given that Mamba can also be used for downstream generation, it is still up to be shown how much compression (if at all) is needed.

---

> ### Author Response · Authors · 2024-11-25
> **Updated manuscript**
>
> Thank you for your comments, we tried to address your concerns and provide an update manuscript, with the sections in red that are altered (we will remove the red before the rebuttal period ends).
>
> ## W2a: The authors do not demonstrate the downstream applicability of 3D structure tokenization
> 3D structure tokenization has been demonstrated to be a useful approach for generative language modeling and discrete diffusion, for example ESM-3 models biomolecular structures entirely with a 4096 codebook tokenizer model. Here, we demonstrate higher reconstruction accuracies than the ESM-3 tokenizer with identical codebook size. The motivation of the 3D structure tokenization approach was not well incorporated in the original submission and we added an additional section to the introduction summarizing previous 3D structure tokenizer work, listing 5+ other works in this domain (lines 51 onwards). Besides generation, structure tokenization has sped up structure search, most famously in FoldSeek (https://www.nature.com/articles/s41587-023-01773-0) .
> We entirely focus on the architectural study as this is, to our knowledge, the first time that selective SSMs are used for quantized auto-encoding. We believe a generative model study is beyond the scope.
>
> ## W2b: bio2token does not reduce the number of atoms, raising doubts about whether it can truly support language model development in the field
> Mamba is conventionally used for language modeling and has been demonstrated to model up to millions of tokens, with plenty of demonstrated examples in biology, e.g. in long-distance modeling of DNA sequences. A downstream generative LLM would indeed require a Mamba-based architecture as well, which should not be a barrier, so we would like to disagree with that comment.
> We agree that sequence length compressibility (congruent to quantization compression) should be investigated, so we now provide a compressibility study (please see main text section 3.2.1, lines 229 and Appendix A.2 table 4). We find that the token sequences are compressible with RMSE increases of 1.7 for a compression factor of 2 and 2.6 for a compression factor of 4, which is in line with compressibility factors reported by others.

---

> ### Comment · Reviewer_rmoF · 2024-11-27
>
> Since the model does not compress the input size, what it essentially does is simply use a 4096-codebook to discretize each coordinate. Based on this reasoning, wouldn't it be possible to divide the 3D space into 4096 grids and discretize coordinates based on the grids they fall into? If such a strategy can achieve results similar to the model, then the model's design seems unnecessarily complicated. My concerns about generalization stem precisely from whether the model merely falls into such trivial solutions implicitly. Furthermore, I still believe that designing a tokenizer is not the ultimate goal. The authors should provide additional experiments to demonstrate the advantages of their tokenizer in practical design tasks.

---

> ### Author Response · Authors · 2024-11-27
> **a learned tokenizer is more efficient than a spatial tesselation**
>
> ### **Codebook Efficiency vs. Spatial Tessellation**
> Section 3.2 of the updated manuscript addresses your question: *"Codebook efficiency: learned tokenizer versus spatial tessellation."* We compare our tokenizer's errors to those of a naive uniform voxelation of space. For instance:
> - A ribosomal RNA with a spatial extent of 100 Å requires 110k voxels to guarantee 1 Å accuracy.
> - Achieving 0.2 Å RMSE for a small molecule of 30 Å extent demands 191k voxels—over an order of magnitude beyond our codebook sizes.
>
> Our method achieves ~0.6 Å accuracy with a 4096-codebook for RNA structures. Appendix A.3 further demonstrates that even a 256-codebook achieves ~1 Å errors for protein structures, highlighting the network's efficiency. To our knowledge, this is the first quantification of spatial compressibility using codebooks.
>
> ---
>
> ### **Input Size Compression**
> Section 3.2, *"Compressibility of tokens"*, details input compression experiments. Token sequences of length N are compressed by factors k \in \{ 1, 2, 4\} , reducing sequence lengths to N / k. Results in Appendix Table 4 show RMSE increases by factors of 1.7 and 2.6 for compression factors of 2 and 4, respectively. These values align with previously reported compressibilities for residue-level tokenizers (*Gaujac et al., 2024*).

---

> ### Comment · Reviewer_rmoF · 2024-11-28
>
> Thank you for the author's response. The new results indeed demonstrate that the model has not fallen into a trivial coordinate discretization scheme. Accordingly, I have raised my score. However, due to the lack of validation on truly meaningful downstream tasks, I still cannot recommend the paper for acceptance.

---

### Official Review · Reviewer_fpX1 · 2024-11-04

**Soundness:** 3
**Presentation:** 3
**Contribution:** 2
**Rating:** 5
**Confidence:** 4

**Summary:**

This work proposes a novel architecture for training quantized auto-encoder of 3D molecular structures. It leverages the mamba architecture and an all atom aligned MSE loss.
The authors perform several trainings of the framework from several datasets, of diverse nature, such as RNA – small molecules – proteins the proposed method achieved competitive reconstruction accuracy compared to established baselines.

**Strengths:**

The authors proposition is straightforward and fairly well motivated for leveraging the mamba architecture.
The proposed models seem effective in the sense that the optimized quantity (i.e.  the all atom RMSE) is low on their test sets. Moreover, the authors proposition seem data efficient notably in the protein setting compared to competitors.

They showcase their proposition in a variety of settings.
Interestingly they show that when training on all data sources at once they do not observe a significant boost or decrease in performances. This seem to illustrate that there is only a little transfer between tasks / datasets given the authors design choices.
The authors also provide an interesting discussion on the limitation of their work.

**Weaknesses:**

**Architecture**

While the authors develop a paragraph dedicated to mamba based SSM, (and since the authors’ proposition heavily relies on the mamba architecture) I would have enjoyed a thorougher description of the design choices, and hyperparameter selection. Indeed, since to the best of my knowledge it is the first work leveraging a SSM deep architecture for 3D structure encoding, it is important for practitioners to understand the rationale behind the design choices.

**Performance comparison**

The authors implement an "all-to-all" atom autoencoding approach, assigning a unique integer code to each atom in a point cloud of NN atoms. This strategy substantially increases the information density compared to other models like ESM-3 or InstaDeep’s quantized autoencoder, which encode only a single integer per residue in protein structures.
While encoding every atom individually this procedure enable (very) fine grain resolution, the authors achieve a much finer level of detail at the expense of a lower compression, therefore I find it difficult to understand the relative advantage of the author’s proposition compared to competitors.

**Invariance**
 To the best of my understanding the authors provide an unprocessed point cloud (centered) suggesting that the rotated point cloud can have a different representation compared to the original one. This remark might require further investigation. Indeed, it could be interesting to understand whether the learned decoder is a surjection.


**POST REBUTTAL**
 I appreciate the authors' response but have decided to maintain my score. While the work is interesting, I find it borderline in the current state and believe it falls short of ICLR's standards for publication.

**Questions:**

1- You report all-to-all RMSD for proteins and only TM on C-alpha, can it be computed all atoms ?

2- When reconstructing proteins how difficult is it to attribute an atom to a residue ?

3- Invariance: As highlighted in the above paragraph, it would be interesting the see if the tokens / output changes when the input point cloud is rotated since the encoding do not seem to be invariant to rotation ? And also what is the reconstruction error distribution of a molecule given a set of rotation.

4 -  Do you expect to obtain significant better results when scaling your datasets ? For instance moving from CATH to pdb or increasing using AF db ?

---

> ### Author Response · Authors · 2024-11-18
>
> Thank you for your review and constructive comments!
>
> ### Q1: “All-atom TM-scores"
> A very reasonable thought, but “all-atom TM-scores" will be distorted if naively calculated out of the box:
> TM-scores are per definition derived from residue-wise alignments (see Equation 1 of the original paper by Zhang et al. https://onlinelibrary.wiley.com/doi/10.1002/prot.20264), conventionally this is done on the C-alpha. Nothing prevents us computationally to perform an alignment and TM-calculation over all atoms, however, this will lead to a distortion of the score. The TM-score formula involves a length scaling factor “d_0” (to guarantee length independence for conventional protein lengths). This factor is empirically derived from a calibration curve for proteins of residue lengths of 10-1000 – see Fig. 3 of the link above. The scaling factor d_0 would require a recalibration for proteins of atom lengths 100-100k. In fact, if you take a closer look at the paper for the RNA TM-score – the d_0 scaling factor is different to the protein d_0 (Fig. 1 here: https://pubmed.ncbi.nlm.nih.gov/31161212/ ). Rerunning the calibration for an “all-atom TM” would be interesting, but we hope the referee agrees that this is a bit off scope.
>
> ### Q2: How to assign atoms to residues
> Protein/RNA residues are ordered by their sequence, and atoms within residues follow the canonical order of backbone (N,Ca, C, O), followed by the side-chain. The side-chain order follows the side-chain net conventions (https://github.com/jonathanking/sidechainnet ). You can find the order in our code (https://anonymous.4open.science/r/bio2token-72F2/bio2token/utils/pdb.py) .
> So if the sequence of the protein/RNA is known, as in our case, it is straightforward. We will add a sentence in the manuscript to explain the canonical ordering used.
>
> ### W3&Q3: Tokens are not rotationally invariant – what does that mean for the token representation and are errors rotationally invariant?
> Very relevant question. We did not investigate to what degree rotational equivariance could enhance performance, we feel it is not needed. We achieve comparable reconstruction errors to the competitor tokenizers with an arguably much simpler approach, and it wasn't expensive to train. We followed the referee’s idea to plot auto-reconstruction errors over full rotations and don’t see any bias. We will update the manuscript with an exemplar analysis, rotating proteins around all axes and plot the errors. In addition, we will add analysis on the local “interaction length” R of a token, i.e. if the atom at position i is changing – how many atoms to the left and right i+/- R are changing. This will hopefully provide some interpretability of the token information on local structure environment and relative orientation. If you have any other ideas for how to provide insight into this token space – let us know!
>
> ### Q4: Training on AFDB
> It was on our to-do list and we are running the trainings – we will report back soon.
>
> ### W2: What is your relative advantage to competitor methods, given the higher information density and lack of compression?
> We don’t have a good answer on what the best modeling resolution is and it will be application dependent (arguably for some applications even lower resolution of k-mer modeling instead of residue-level might be better, if one wants to model biomolecular interactions -- atom level will likely be the most efficient approach).
> Here, our competitive advantage is two-fold:
> 1. **Molecule-class independence**. Atomistic resolution opens up the entire array of biomolecular classes. To our knowledge, none of the competitor models can encode and decode RNA structures out of the box.
> 2. **High information density**: Atom-level modeling is computationally prohibitive with IPA, so we think it is worth the investigation, with mamba we can "afford" to not compress and learn at high density. Can the referee clarify why he regards this as a disadvantage?
>
> We definitely agree, a proper comparison of computational efficiency and performance between Mamba and IPA approaches is needed. We are currently running comparisons on small molecules of Mamba versus IPA training (all-atom proteins are infeasible with IPA). We will report back to give a more numerically informed answer. We also agree that compressibility should be investigated and will report back with a manuscript update over the course of the week.

---

> ### Author Response · Authors · 2024-11-25
> **Follow up: Results on invariance and Alphafold DB training**
>
> We see the reviewer has supplied a "post-rebuttal" response before we were able to upload their requested changes. In addition to our previous response, we now provide the results to your questions. Please refer to the updated manuscript, sections with major alterations are in red (which we will change before the final rebuttal deadline).
>
> ## W1: Please provide more details on the hyperparameters and architecture choices
> We now provide a through investigation of model size, codebook size and efficiency, as well as a comparison study to invariant point attention. Please take a look at the new sections 3.2, particularly 3.2.1. We further provide an ablation study of the final model. (section 3.2.2 in the new version)
>
> ## Q3a: Invariance: Do tokens change when the input point cloud is rotated?
>
> Tokens are rotational variant and change in a circular fashion -- the amount of orientation change of an atom, with respect to the coordinate space centre, is reflected in the token changes. In the updated manuscript we provide Figure 4 as a case study how the atom tokens of an exemplar amino acid change with respect to rotations of the protein. We provide more detailed studies on token interpretability, including a study on token mixing radius (how many atoms influence the token at a given position?), depending on the number of encoder blocks, where we show that is approximately linear. This can be found in Appendix A2. We suspect the reviewer might find it interesting, given their particular questions on the token interpretability.
> ## Q3b: What is the reconstruction error distribution of a molecule given a set of rotation?
>
> It is uniform, there is no orientation bias. This is also due to rotation augmentation leveraged in training, which we did not make clear in the text previously (added now). We provide an exemplar error distribution plot for a set of rotations in the Appendix A.5.2, Figure 9.
>
> ## Q4: Do you expect better results with more data, e.g. scaling to AFDB
> Yes! This very much the case and we followed the suggestion and trained with an additional subset of 100,000 Alphafold DB proteins, using FoldSeek clusters. Our training results did improve by a fair amount. Please refer to the manuscript for all updated tables. We see RMSE improvements by about 0.2 Angstrom for bio2token and protein2token, see Table 2 in the main text.

---

> > ### Author Response · Authors · 2024-11-30
> >
> > We would like to enquire if the reviewer had the opportunity to review the additional material and analysis provided in the updated manuscript in response to their comments. We think we addressed many of the referee's concerns, including training on new data, and additional analysis, as they requested. We'd appreciate to hear your feedback.

---

### Official Review · Reviewer_gfkK · 2024-11-08

**Soundness:** 3
**Presentation:** 3
**Contribution:** 3
**Rating:** 6
**Confidence:** 3

**Summary:**

This paper proposes to train a mamba-based auto-encoder on biomolecular structures to allow for accurate tokenization (i.e. conversion to discrete tokens). The authors compare training several domain-specific tokenizers vs one shared one, and investigate scalability.

**Strengths:**

The modelling choices are generally sound and practical, with the choice to go for mamba over transformers well-justified for the domain. Bio2token and other tokenizers proposed in this work appear to be highly scalable, allowing for an all-atom representation to be used for a range of chemical objects.

**Weaknesses:**

From reading the paper, I am not sure how useful the tokenizer is in itself, and how exactly it enables new models that could build on top of it. Would the main downstream models making use of the pretrained tokenizer be generative or predictive in nature? Is the tokenizer at all useful on its own? Moreover, I am wondering how can we know the models that build on top of bio2token would be useful in the face of compounding errors (i.e. errors stemming from the tokenizer itself adding up with errors of the downstream model)?

=== Update 02/12/2024 ===

During the discussion period the authors have argued that structure tokenizers such as the one proposed in this work are widely used in the field, for example by generative models. While they do not present results showing downstream improvements, they do compare with other tokenizers which were already evaluated in downstream tasks, so it is reasonable to expect the improved tokenizer would also lead to improvements downstream, even though one can't be sure. The authors also included several new results and expanded the discussion. To reflect this, I raise my score. However, I leave my confidence as low, as it's not clear to me how confident we can be that the improved tokenization leads to improvements in downstream tasks without testing it directly.

**Questions:**

See the "Weaknesses" section above for specific questions.

---

> ### Author Response · Authors · 2024-11-25
>
> Thank you for your comments.
> All changes in the manuscript are marked in red.
>
> ## Q1: Are the tokens useful?
>
> 3D structure tokenization has been demonstrated to be a useful approach for generative language modeling and discrete diffusion, for example ESM-3 models biomolecular structures entirely with a 4096 codebook tokenizer model. Here, we demonstrate higher reconstruction accuracies than the ESM-3 tokenizer with identical codebook size. The motivation of the 3D structure tokenization approach was not well incorporated in the original submission and we added an additional section to the introduction summarizing previous 3D structure tokenizer work, listing 5+ other works in this domain (lines 51 onwards). Besides generation, structure tokenization has sped up structure search, most famously in FoldSeek (https://www.nature.com/articles/s41587-023-01773-0) .
>
> ## Q2: I am worried about compounding errors in downstream models
> Errors should not be compounding, but the tokenizer accuracy will provide a glass-ceiling RMSE for any downstream model. We would like to emphasize that 3D structure tokenizers are employed by generative models like ESM-3. The ESM-3 tokenizer model has worse RMSE than our all-atom bio2token with the same codebook size.
>
> Our model is (to our knowledge) the first to deploy a selective SSM/Mamba in an auto-encoder to encode all-atom structures. We fully focus on a thorough study of architecture, it's efficiency and performance. In response to other reviewers, we now provide detailed architectural studies into model sizes, codebook size, a computational efficiency comparison between Mamba and invariant point attention and token sequence compressibility (please see the revised manuscript, section 3.2 onwards).
> We hope the reviewer nonetheless find this study valuable, however, we do understand their concern that we did not demonstrate generation, which we found to be beyond the scope.

---

> > ### Author Response · Authors · 2024-11-30
> >
> > Given that the discussion period ends in two days, we would like to enquire if the reviewer's concerns are addressed by the changes and additional material in the revised manuscript, or if any questions remain. We are looking forward to your comments.

---

### Meta-Review · Area_Chair_Bozw · 2024-12-21

**Metareview:**

(a) The paper proposes a method for all-atom tokenization of biomolecular structures using a quantized auto-encoder with the Mamba state space model. It claims to achieve high reconstruction accuracies for proteins, RNA, and small molecules. The method is shown to be scalable to large systems and more efficient than some existing approaches.

(b) Strengths:
  - The use of Mamba architecture for efficient all-atom modeling is a new approach.
  - The provided comparisons with other tokenizers and the demonstration of improved reconstruction accuracies in some cases are valuable.

(c) Weaknesses:
  - Lack of clear demonstration of the practical utility of the tokenizer in downstream applications.
  - Incomplete evaluation of certain aspects such as rotational invariance and computational efficiency in the initial submission.
  - Some details about the model architecture and hyperparameters were initially lacking.

(d) Reasons for Rejection:
  - Despite improvements in the rebuttal, the lack of direct validation in truly meaningful downstream tasks remains a major concern. The paper focuses mainly on the architectural study and reconstruction accuracy, but it is not clear how the proposed tokenizer will be effectively used in practical applications such as generative models.
  - While the authors addressed many of the reviewers' concerns, the overall contribution may not be sufficient for acceptance at ICLR. The paper does not convincingly show that the proposed method has a significant impact on the field beyond what is currently available.

**Additional Comments On Reviewer Discussion:**

(a) Reviewer Points and Author Responses:
  - Downstream Applicability: Reviewers questioned the usefulness of the tokenizer in downstream applications. Authors provided examples of how 3D structure tokenization has been useful in generative language modeling and structure search (e.g., ESM-3 and FoldSeek), but did not conduct downstream experiments.
  - Model Details: Concerns were raised about the lack of detail in the architecture description and hyperparameter selection. Authors added more details on the architecture, conducted ablation studies, and provided comparisons with other methods (e.g., IPA).
  - Evaluation Metrics: Reviewers asked for additional evaluation metrics such as all-atom TM-scores and chemical correctness. Authors explained the challenges with all-atom TM-scores and provided some chemical validity analysis for small molecules.
  - Compressibility and Generalization: Questions were raised about the lack of compressibility and the generalization ability of the model. Authors conducted compressibility experiments and addressed the generalization concerns by showing that the tokenizer can capture spatial conformations well.

  (b) Weighing the Points:
The authors' responses improved the paper in many aspects, but the lack of direct downstream application validation was a crucial factor in the final decision. While the improvements in model details and evaluation metrics were valuable, they did not fully compensate for the lack of a clear practical impact.

---

### Decision · Program_Chairs · 2025-01-22

Reject